# Survivorship in Colorectal Cancer: A Cohort Study of the Patterns and Documented Content of Follow-Up Visits

**DOI:** 10.3390/jcm9092725

**Published:** 2020-08-24

**Authors:** Victoria Garwood, Karolina Lisy, Michael Jefford

**Affiliations:** 1Department of Cancer Experiences Research, Peter MacCallum Cancer Centre, Melbourne, VIC 3000, Australia; victoria.garwood@mh.org.au (V.G.); Karolina.Lisy@petermac.org (K.L.); 2Faculty of Medicine, Dentistry and Health Sciences, University of Melbourne, Parkville, VIC 3010, Australia; 3Australian Cancer Survivorship Centre, Peter MacCallum Cancer Centre, Melbourne, VIC 3000, Australia; 4Sir Peter MacCallum Department of Oncology, University of Melbourne, Parkville, VIC 3010, Australia

**Keywords:** cancer survivors, aftercare, guideline adherence, cancer survivorship

## Abstract

Survivors of colorectal cancer (CRC) may experience a range of physical, psychosocial, and practical challenges as a consequence of their diagnosis. We assessed the patterns and documented content of follow-up visits within the first three years following treatment, in comparison to survivorship care guidelines. Survivors with stage I-III CRC who underwent curative resection at Peter MacCallum Cancer Centre from July 2015 to January 2018 were followed for up to 1080 days. Patterns of follow-up were calculated by recording the date and specialty of each visit; documented content was assessed using a study-specific audit tool for the first year (360 days) of follow-up. Forty-eight survivors comprised the study population, 34 of whom (71%) attended the recommended two to four follow-up visits in their first year. Visit notes documented new symptoms (96%), physical changes (85%), physical examination (63%), and investigations (56%–90%); none had documented discussions of screening for other primary cancers, or regular health checks and/or screening. Each survivor had at least one outpatient letter that was sent to their primary care physician, but responsibilities were not adequately defined (31%). Although survivors had regular follow-up in their first year, documentation did not consistently address aspects of wider survivorship care.

## 1. Introduction

In 2020, colorectal cancer (CRC) is predicted to be the fourth most commonly diagnosed cancer in Australia [1]. The majority of Australians are diagnosed with early stage (I-III) CRC, with 5 year relative survival of CRC increasing by 18% over a 29-year period, from 1986–1990 to 2011–2015 [2]. Improvements to early detection and treatment have contributed to the expanding population of survivors worldwide [2,3,4]. In the United States, there were a predicted 1.5 million people living with a history of CRC in 2019, which is predicted to rise to more than 1.9 million in 2030 [4].

Cancer survivors may face a multitude of physical, psychological, social, and practical challenges as a consequence of their diagnosis and treatments [5,6]. These can be experienced during active treatment, but also in the weeks, months, and years following treatment completion [5,7]. To address the needs of this growing survivorship population, the Institute of Medicine produced the landmark publication “From Cancer Patient to Cancer Survivor” [5]. For CRC, the first comprehensive practice guidelines were released by the American Cancer Society (ACS) in 2015 [8]. These recommendations relate to the first five years following treatment, broadly addressing four aspects of survivorship care: (1) history and examination in the form of follow-up visits; (2) post-treatment surveillance; (3) wider survivorship care; and (4) care coordination, in particular with primary care. In the first two years following treatment, survivors are advised to have two to four follow-up visits each year. From years three to five, two visits per year are recommended. Post-treatment surveillance incorporates a combination of colonoscopy and/or sigmoidoscopy, carcinoembryonic antigen (CEA) testing and computed tomography (CT) of the chest, abdomen, and pelvis. Wider survivorship care is inclusive of screening for other primary cancers, consideration of long-term and late effects, and lifestyle counselling, as well as general health checks and screening as for the general population [8]. These guidelines are in line with others published by leading institutions, although the timing, frequency, and modality of follow-up varies or is not stipulated [9,10,11,12,13,14].

Although there is increasing recognition of the needs of CRC survivors, implementation of guidelines into practice is unclear [8,15]. Previous literature has focused on the receipt of post-treatment surveillance and office visits, demonstrating wide variation in overall adherence from 12–87% [16].

There is a need to understand the current practices of CRC survivorship care, particularly with regards to follow-up visits conducted by specialists in tertiary care. To our knowledge, there has been no review of the content of follow-up visits for CRC survivors, inclusive of wider survivorship considerations and coordination with primary care. The aims of this study were to determine: (1) what are the patterns of follow-up visits in CRC survivors in the first three years post-treatment, compared to the frequency recommended by guidelines; and (2) to what extent is the documented content of these visits concordant with recommended guidelines?

## 2. Methods

This cohort study was undertaken at Peter MacCallum Cancer Centre (Peter Mac), a comprehensive cancer centre in Melbourne, Victoria, Australia. The design was guided by the Strengthening the Reporting of Observational Studies in Epidemiology (STROBE) statement [17]. Ethical approval was sought from the Human Research Ethics Committees at Peter Mac and was approved on 5 February 2019. Waiver consent was approved by the institutional HREC, given the retrospective nature of the study and the use of medical records.

### 2.1. Participants

Health administrative data was used to compile a list of survivors with an inpatient admission of any kind at Peter Mac between July 2015 and January 2018, who had a diagnosis of non-metastatic CRC without an additional primary cancer, according to the relevant tenth revision of the International Statistical Classification of Diseases and Related Health Problems (ICD-10) [18]. Using medical records, survivors were deemed eligible if they had a histological diagnosis of Stage I-III CRC according to the seventh edition of the American Joint Committee on Cancer (AJCC) Cancer Staging Manual [19] and underwent curative resection at Peter Mac between July 2015 and January 2018. Disease staging for survivors who underwent neoadjuvant therapy was based on initial (pre-treatment) magnetic resonance imaging (MRI). Survivors were excluded if they had a diagnosis of goblet cell carcinoma, asynchronous cancer, familial cancer (familial adenomatous polyposis, hereditary non-polyposis CRC), a previous primary cancer, or inflammatory bowel disease, as follow-up of these survivors was unlikely to be reflective of typical practice. Survivors who did not have their follow-up visits at Peter Mac, were on a clinical trial mandating conditions of their follow-up, had recurrence of their disease prior to the start of the follow-up period, or did not have at least 360 days of follow-up were also excluded. Peter Mac is a comprehensive cancer centre that is quaternary in nature. Patients are often referred to the colorectal cancer (CRC) surgical service to manage patients with recurrent rectal cancer and peritoneal metastatic disease. Compared to many CRC clinical services, Peter Mac sees few patients with de novo CRC, treated with curative intent. Therefore, our study sought to identify a subpopulation of patients whose treatment was completed in entirety at Peter Mac, in order to capture all of their follow-up visits during the study period.

### 2.2. Period of Follow-Up

The beginning of the follow-up period (index date) commenced 90 days post-treatment, defined as the final date of curative treatment, inclusive of surgery and adjuvant therapy. This period was selected to exclude outpatient visits unrelated to survivorship care. Previous sensitivity analyses have not demonstrated any difference in outcomes by extending this period to 6 or 12 months following curative resection [20]. The study population was followed for up to a period of 1080 days, defined as 1–360 days for the first follow-up year, 361–720 days for the second year, and 721–1080 days for the third year. Follow-up visits were defined as outpatient appointments that were attended by survivors at Peter Mac and were conducted by medical oncology, radiation oncology, or surgical oncology.

### 2.3. Data Collection and Handling

Study data was managed using REDCap electronic data capture tools hosted at Peter Mac [21]. The following baseline data were obtained from the medical records of each survivor: (1) Sociodemographic characteristics: age, postcode, sex, smoking status at curative resection; and (2) Disease characteristics: year of curative resection, cancer grade, cancer site, cancer stage, cancer type, receipt of neoadjuvant therapy, and receipt of adjuvant therapy.

The medical records of each survivor were used to record the date (in days since the index date), and medical specialty of each follow-up visit, grouped by follow-up year. The documentation of each visit (e.g., letters and progress notes) were assessed using a study-specific audit tool developed by the authors (see Appendix A) outlining 14 elements of survivorship care, based on recommendations of the ACS [8]. Each element was dichotomised as yes/no according to each follow-up visit by one of the authors (V.G.) and later grouped by follow-up year as an annual yes/no score, with an annual “yes” score defined as at least one follow-up visit that year having adequately addressed the criterion.

Perusal of preliminary data revealed that only a small proportion of survivors had been followed up for the entire three-year period, hence the decision was made to limit analysis of the documented content of visits to the first follow-up year only.

### 2.4. Statistical Considerations

Descriptive statistics were computed for baseline characteristics and the primary outcomes, using R version 3.6.0, manufactured by R Foundation for Statistical Computing in Vienna, Austria [22] and RStudio version 1.2.1335, manufactured by RStudio, Inc. in Boston, MA, USA [23], whilst figures were created with the additional use of the ggplot2 and extrafont packages [24,25]. Given the small sample size, inferential statistics were not calculated.

## 3. Results

### 3.1. Participants

Two-hundred and six patients were identified through health administrative data and screened for inclusion into the study population (Figure 1). One-hundred and fifty-eight survivors were excluded, most commonly as they did not have curative resection during their inpatient admission between July 2015 and January 2018 (*n* = 44, 28%). The study population comprised 48 survivors, for whom the patterns of follow-up visits were determined, and the documented content of their follow-up visits assessed. Analysis was restricted to the first 360 days of follow-up.

### 3.2. Survivor Characteristics

The characteristics of the study population are summarised in Table 1. The age of survivors varied from 28 to 88 years, with a median age of 59 years. Twenty-seven survivors (56%) were female; the majority of survivors were non-smokers (*n* = 29, 60%), and lived in a major city (*n* = 29, 60%). The majority of survivors were diagnosed with colon cancer (*n* = 23, 48%), whilst fifteen survivors had rectal cancer (31%), and ten survivors had rectosigmoid cancer (21%). AJCC staging varied from I-IIIC, with most survivors diagnosed at stage III (*n* = 19, 40%), and stage II (*n* = 18, 38%). Receipt of treatment varied by cancer site; the most frequent combination in cancer of the colon (*n* = 13, 57%) and rectosigmoid (*n* = 5, 50%) was surgery alone, whilst most rectal cancer survivors had neoadjuvant chemoradiotherapy, surgery, and adjuvant chemotherapy (*n* = 7, 47%).

### 3.3. Patterns of Follow-Up Visits

Patterns of follow-up for the study population for the three-year period of 1080 days are depicted in Figure 2. The follow-up period of survivors varied from 367 to 1080 days. Mean follow-up was 661 days (SD 269), and median follow-up was 576 days (IQR 408, 969). Only six survivors (13%) were followed for the entire three-year (1080 days) duration. The majority of survivors were censored due to study end date (*n* = 39, 81%). Both the timing and frequency of visits varied amongst survivors. In survivors who attended more than one specialty, it was variable as to whether these visits were timed in concert.

#### 3.3.1. Patterns of Follow-Up Visits in the First Year

One-hundred and eighty-seven visits were attended by the 48 survivors in their first year of follow-up, with the majority delivered by surgical oncology (*n* = 166, 89%), followed by medical oncology (*n* = 20, 11%) and radiation oncology (*n* = 1, 5%). The range of visits attended by each survivor varied from one to eight; 34 survivors (71%) had two to four visits, in line with the ACS guidelines (Figure 3). Two survivors (4%) had a single visit in their first year of follow-up, whilst 12 survivors (25%) had five or more visits at a rate higher than recommended by the ACS guidelines.

In Appendix A, we outline the patterns of follow-up visits in the first year stratified by treatment (surgery only, receipt of neoadjuvant and/or adjuvant therapy). With respect to survivors who had surgery only, 17 survivors (81%) had two to four visits, 3 survivors (14%) had five visits or more, and 1 survivor (5%) had one visit. In the group that had neoadjuvant and/or adjuvant therapy, there were a higher proportion of survivors that attended more than four visits (*n* = 9, 33%).

#### 3.3.2. Patterns of Follow-Up Visits in the Second Year

Fifty-eight visits were attended by 18 survivors in their second year of follow-up, with the majority delivered by surgical oncology (*n* = 55, 95%), followed by medical oncology (*n* = 3, 5%). The range of visits attended by each survivor varied from one to six; 13 survivors (72%) had two to four visits, in line with the ACS guidelines (Figure 4). Two survivors (11%) had a single visit in their second year of follow-up, whilst three survivors (17%) had five or more visits at a rate higher than recommended by the ACS guidelines.

#### 3.3.3. Patterns of Follow-Up Visits in the Third Year

Twelve visits were attended by six survivors in their third year of follow-up, all of which were delivered by surgical oncology. The range of visits varied from one to three; one-third (*n* = 2) of survivors had two visits, in line with ACS guidelines. The other two-thirds of survivors were equally divided between one and three visits.

### 3.4. Documented Content of Follow-Up Visits in the First Year

Figure 4 summarises the number of survivors in the study population who had at least one follow-up visit in their first year (360 days) that documented aspects of the 14 elements of survivorship care, according to the study-specific audit tool (Appendix A). In at least one visit, 46 survivors (96%) had appropriate documentation of new symptoms, and 30 survivors (63%) had documentation of a physical examination. The majority of survivors had a documented discussion of physical changes (*n* = 41, 85%). With regards to post-treatment surveillance, the majority of survivors had relevant documentation of results for colonoscopy (*n* = 27, 56%), CT (*n* = 37, 77%), and CEA (*n* = 43, 90%). Forty-three (90%) survivors had any additional investigations appropriately justified, for example positron electron tomography (PET) if there were concerns of recurrence. With the exception of discussion of physical changes, wider survivorship care was poorly addressed. None of the survivors had documented discussions of screening for other primary cancers, or the need for regular health checks and/or screening. Only four survivors (8%) had documented discussions of psychological changes, whilst less than half of the survivors had documentation relevant to discussions of social and/or practical changes (*n* = 15, 31%), or lifestyle behaviours (*n* = 22, 46%). All survivors had at least one follow-up visit containing evidence of an outpatient letter to a primary care physician (PCP) and/or another specialist, although only 15 survivors (31%) had documentation that designated roles and/or responsibilities to others involved in their care.

In Appendix A, we outline the documented content of follow-up visits in the first year stratified by treatment (surgery only, receipt of neoadjuvant and/or adjuvant therapy). Survivors who underwent neoadjuvant and/or adjuvant therapy in addition to surgery had higher adherence to survivorship care elements, with the exception of additional investigations being appropriately justified. When organised by year of curative resection, there was no apparent trend in improved documentation over the observational period.

## 4. Discussion

This study examined the concordance between recommended survivorship care, and the actual care of CRC survivors. Although most survivors attended the recommended two to four visits in their first and second year of follow-up, many had a greater number of visits. This appeared to be more common in survivors who had neoadjuvant and/or adjuvant therapy, in addition to curative resection. Documentation consistently addressed aspects of post-treatment surveillance, as well as history and examination. Although every survivor had at least one outpatient letter sent to their PCP, most did not have any documented discussions of wider survivorship care, beyond physical changes. These aspects were more likely to be addressed by survivors who had neoadjuvant and/or adjuvant therapy, in addition to curative resection.

Our results draw comparisons to the growing body of literature regarding adherence to follow-up visits in CRC survivors. Observational studies published in the past decade have indicated that the proportion of survivors who attended at least two visits in their first year post-treatment varied from 61–98% [27,28,29,30,31,32,33]. In one investigation, approximately 19% of survivors had more than five visits, broadly similar to our own data [33].

Notwithstanding the availability of survivorship care guidelines that encourage a holistic approach to survivorship care, it appears that current practice remains focused on post-treatment surveillance. Detection of recurrent disease has been identified as a primary aim of follow-up amongst patients, carers, specialist nurses, and physicians; however, survivors and carers tend to have higher expectations and perceived benefits from investigations [34]. Nonetheless, a recent systematic review and meta-analysis found no overall survival benefit for intensive surveillance [35]. In our study, the overwhelming majority of follow-up visits were conducted by surgical oncology (*n* = 233, 91%). This is in contrast to other observational studies, in which medical oncologists are commonly designated as the primary cancer specialist [27].

Results from this study suggest that several recommended aspects of wider survivorship care are lacking. This is important, given that many considerations, such as lifestyle behaviours, have been associated with reduced risk of cancer recurrence, other cancers, and other illnesses [8]. A recent cross-sectional survey of CRC survivors in Australia at one year post-diagnosis found that 24% reported having a discussion with a physician about screening for other primary cancers; and that 30% had received assistance with lifestyle changes [36]. We also noted infrequent documentation of the role of various providers, in particular the PCP. This is likely a contributing factor to the discrepancies amongst specialists and PCPs regarding perceived roles in follow-up care [37]. Specialists express a desire to remain involved in survivorship care, but uncertainty in the capabilities of the PCP [37]. This is despite PCPs believing that their scope of practice can facilitate preventative medicine, enhance psychosocial care, and improve health literacy and psychosocial care [37]. This scope of practice could help to address unmet needs of the cancer survivor, with physical, financial, and educational needs the most frequent [38]. In a recent systematic review of cancer survivors in Australia, assistance with psychosocial issues ranked as the most prevalent unmet need [39]. Without adequate coordination amongst providers, care can be repetitive, expensive, lacking in evidence, or not patient-centred [15]. This was evidenced in the potential overuse of visits in our cohort throughout the three follow-up years, which was higher in those known to more than one specialty. It should be noted, however, that not all additional visits should be considered overuse; many of our patients who attended a greater number of visits were being closely monitored for a persistent physical or psychological symptom, for example.

There are significant challenges to delivering optimal survivorship care, including the growing number of survivors and limited health workforce [5,15,40]. Alternative models of care have been considered, which could include the greater use of formalised models of shared care [40,41]. Other strategies to improve guideline adherence might include providing survivors with a question prompt list to direct the content of consultations, use of patient-reported outcomes, and note templates contained within electronic medical records. Efforts to improve quality reporting may also facilitate greater adherence to recommended survivorship care [6,15].

### Strengths and Limitations

A number of limitations should be considered in the interpretation of our results. A previous systematic review showed that the completeness of diagnostic reporting with ICD-coded data had a mean sensitivity of approximately 67% [42]. This may have meant that survivors who were eligible were not included in the original dataset obtained from health administrative data. When screening survivors for inclusion into our study, we did find that a significant proportion of survivors had inaccurate coding that led to their exclusion. Although documentation in our study was more complete for investigations than for other survivorship aspects, there are limitations in making inferences from the medical records alone. It is certainly possible that discussions did take place but were not documented. The criteria used in our study-specific audit tool (Appendix A), however, were generous in terms of the content that was deemed as appropriate documentation. In addition, survivors who had more visits had more opportunities to address the survivorship care criteria in our audit tool. This could account for the differences noted between survivors who had surgery only, and those receiving neoadjuvant and/or adjuvant therapy, who were more likely to attend more visits. Finally, our small sample size and the heterogeneity of our survivorship cohort, brings into question the generalisability of our findings. Our data warrant further exploration to ascertain whether these findings are also seen in other services. Nonetheless, our results provide novel insights into follow-up at a comprehensive cancer centre in an Australian setting, suggesting that documentation is inadequate, and that practice has not kept pace with the provision of survivorship care guidelines. The identification of aspects of clinical practice that are poorly documented will provide avenues for future research, with the aim to improve and better coordinate care.

## 5. Conclusions

Although survivors regularly attended visits in their first year of follow up in line with recommended guidelines, documentation was inconsistent across many aspects of survivorship care, with many key areas poorly documented. These results contribute to a growing body of evidence that suggests practice has remained focused on detection of disease recurrence and has not adequately expanded to consider the breadth of care as reflected in survivorship care guidelines. Future research should focus on strategies to enable guideline-concordant follow-up care, recognising that oncology specialists require education and training to address the many unmet needs of the cancer survivor.

## Figures and Tables

**Figure 1 jcm-09-02725-f001:**
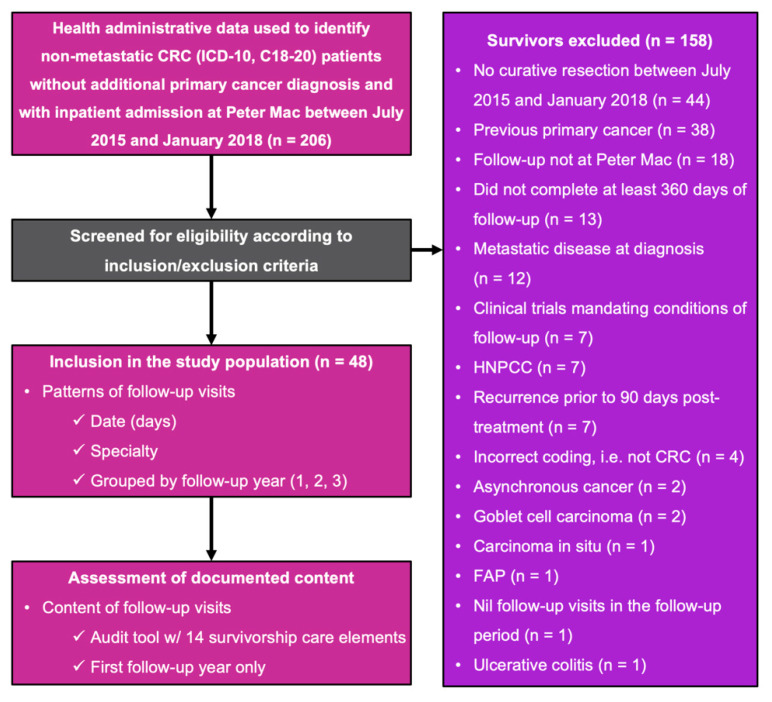
Flow of participants through the study. Two-hundred and six patients were screened for eligibility. Forty-eight survivors, whom had at least 360 days of follow-up, had their patterns of follow-up determined and their documented content assessed for each visit using a study-specific audit tool (Appendix A). CRC: colorectal cancer; FAP: familial adenomatous polyposis; HNPCC: hereditary nonpolyposis colorectal cancer; ICD-10: tenth edition of the International Statistical Classification of Diseases and Related Health Problems; Peter Mac: Peter MacCallum Cancer Centre.

**Figure 2 jcm-09-02725-f002:**
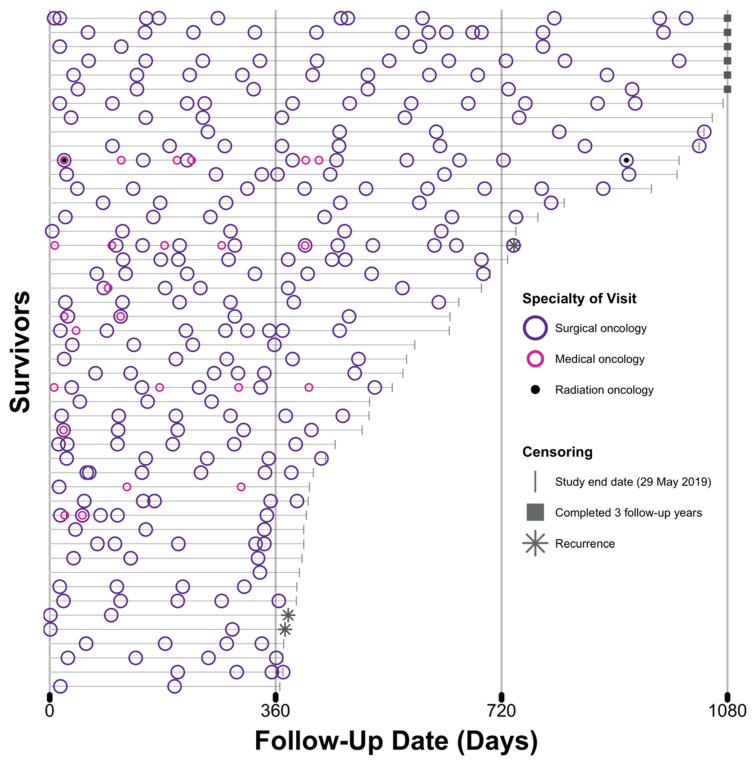
Patterns of follow-up visits in the study population over the observational period (*n* = 48). Follow-up visits are represented by circles corresponding to the specialties of surgical oncology (purple), medical oncology (pink) and radiation oncology (black). Each grey horizontal line represents the follow-up period (in days) of an individual colorectal cancer survivor until censoring, as represented by a grey line (study end date), square (completion of three years of follow-up), and star (recurrence).

**Figure 3 jcm-09-02725-f003:**
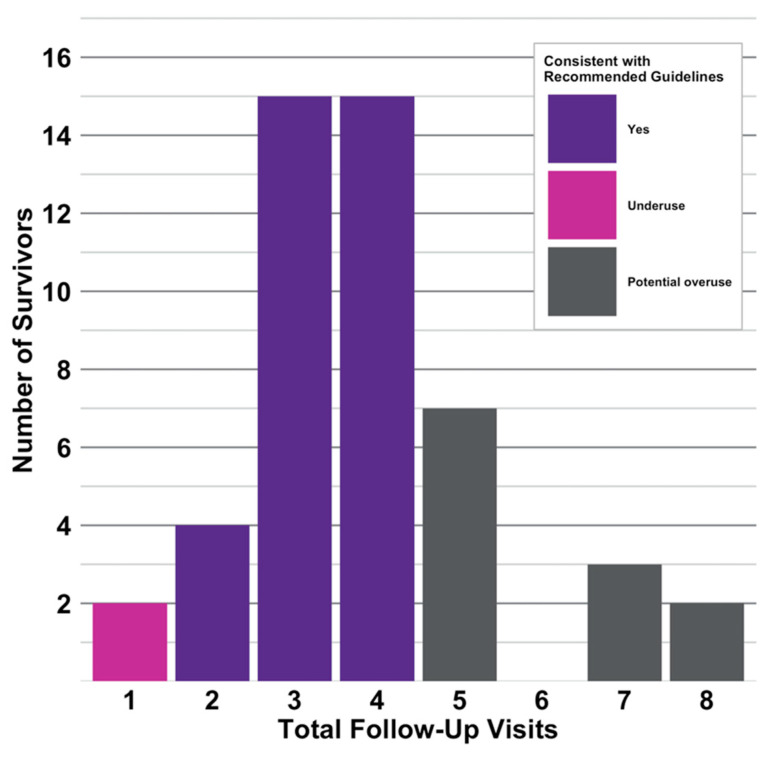
Frequency of follow-up visits in the study population (*n* = 48) in the first year of follow-up (360 days). The colour of the columns indicates whether the number of visits was consistent with the ACS survivorship care guidelines [8]. ACS: American Cancer Society.

**Figure 4 jcm-09-02725-f004:**
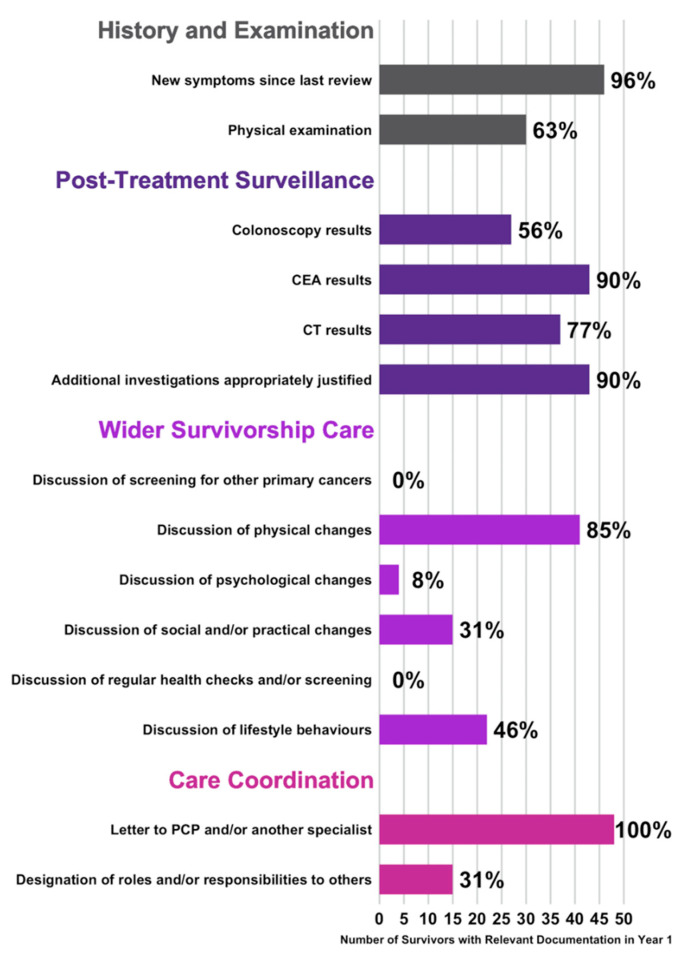
Documented content of follow-up visits in the study population (*n* = 48) in the first year of follow-up (360 days). The x-axis denotes the number of survivors that had at least one visit in their first year containing appropriate documentation relevant to the specific element of survivorship care according to the ACS survivorship care guidelines [8]. These are broadly represented by four key aspects of history and examination (grey), post-treatment surveillance (dark purple), wider survivorship care (magenta), and care coordination (pink). Percentages are denoted at the end of the bar graphs. ACS: American Cancer Society; CEA: carcinoembryonic antigen; CT: computed tomography; PCP: primary care physician.

**Table 1 jcm-09-02725-t001:** Sociodemographic and Disease Characteristics of the Study Population (*n* = 48).

**Sociodemographic**		
**Age at curative resection (yrs)**		
Mean [SD]		58.9 [14.8]
Median [IQR]		59 [50, 69]
Min, Max		28, 88
**Remoteness ^a^**		***n* (%)**
RA1 (Major Cities of Australia)		29 (60.4%)
RA2 (Inner Regional Australia)		14 (29.2%)
RA3 (Outer Regional Australia)		5 (10.4%)
**Sex**		***n* (%)**
Female		27 (56.3%)
Male		21 (43.8%)
**Smoking status**		***n* (%)**
Non-smoker		29 (60.4%)
Ex-smoker		11 (22.9%)
Current smoker		8 (16.7%)
**Disease**		
**Cancer site and treatment**		***n* (%)**
Colon		23 (47.9%)
	Surgery only	13 (56.5%)
	Surgery and adjuvant chemotherapy	9 (39.1%)
	Neoadjuvant chemotherapy, surgery and adjuvant chemotherapy	1 (4.3%)
Rectal		15 (31.3%)
	Neoadjuvant chemoradiotherapy, surgery and adjuvant chemotherapy	7 (46.7%)
	Surgery only	3 (20%)
	Neoadjuvant chemoradiotherapy and surgery	2 (13.3%)
	Neoadjuvant chemoradiotherapy, surgery and adjuvant chemoradiotherapy	1 (6.7%)
	Neoadjuvant radiotherapy and surgery	1 (6.7%)
	Surgery and adjuvant chemotherapy	1 (6.7%)
Rectosigmoid		10 (20.8%)
	Surgery only	5 (50%)
	Neoadjuvant chemoradiotherapy, surgery and adjuvant chemotherapy	2 (20%)
	Surgery and adjuvant chemotherapy	2 (20%)
	Neoadjuvant chemoradiotherapy and surgery	1 (10%)
**Year of curative resection**		***n* (%)**
2015		7 (14.6%)
2016		13 (27.1%)
2017		20 (41.7%)
2018		8 (16.7%)
**Cancer grade**		***n* (%)**
Low		41 (85.4%)
High		7 (14.6%)
**Cancer stage ^b^**		***n* (%)**
I		11 (22.9%)
IIA		12 (25%)
IIB		1 (2.1%)
IIC		5 (10.4%)
IIIA		1 (2.1%)
IIIB		12 (25%)
IIIC		6 (12.5%)
**Cancer type**		***n* (%)**
Adenocarcinoma (NOS)		36 (75%)
Adenocarcinoma (Usual Type)		6 (12.5%)
Adenocarcinoma (Mucinous)		5 (10.4%)
Adenocarcinoma (Colloid)		1 (2.1%)

Abbreviations. AJCC: American Joint Committee on Cancer; IQR: interquartile range; Max: maximum; Min: minimum; MRI: magnetic resonance imaging; NOS: not otherwise specified; SD: standard deviation. ^a^ As determined by the Australian Statistical Geography Standard (ASGS) Volume Five using survivor postcodes [26]. ^b^ Classified according to the seventh edition of the AJCC Cancer Staging Manual on histology [19], or initial (pre-treatment) MRI (receipt of neoadjuvant therapy).

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
