# Peer review of "Survivorship in Colorectal Cancer: A Cohort Study of the Patterns and Documented Content of Follow-Up Visits"

_jcm, 2020, doi:10.3390/jcm9092725_

Round 1

Reviewer 1 Report

Thank you for giving me the opportunity to review this manuscript. Cancer survivorship is an important part of the cancer journey, yet, it is unstructured and is poorly coordinated. 

The study by Garwood et al. evaluated the frequency, content, and coordination of care after completion of active therapy in colorectal cancer patients diagnosed between 2015-2018.

The authors show that the majority of the visits are performed by surgeons, this is not unexpected as most patients in this study had surgery only. It would be valuable to show the patterns of survivorship for patients who had surgery only vs. those who had surgery + other modalities of treatment (such as chemotherapy and radiation). 

The authors state that the documentation in most cases didn't address the cancer screening issues. The age of the study population is 28-88. Most cancer screening have an upper limit of age. Therefore a significant proportion of these patients may not be a candidate for screening. The authors should report the cancer screening for patients who are diagnosed before their 74th birthday for addressing screening in cancer survivors.

Is there a trend for better content over the study period?

The study population is small and very heterogenous and caution should be exercised in drawing conclusions. Concluding that primary care physicians can be part of survivorship is not supported by the data presented in this manuscript. Primary care physicians need to be informed about the survivorship care plan and need adequate training on the issues related to survivorship, I don't think they can be responsible for survivorship care when surgeons, medical oncologists, and radiation oncologists, who do cancer care everyday are not clear on the required tasks for survivorship care.

Reviewer 2 Report

Authors present a cohort of follow-up survival visits from their institution. During the 2.5 years of study, they include 48 pts, so <20 a year. This small volume per year (not the confounding, as they suggest in the limitations) is a major reason that raises concerns about the generalizability about the results. 

Some other major concerns:
1) the authors exclude 75% of their cancer patients -they claim that the reasons are 'not reflective of typical practice' but I would make the case if you are limiting to only 1 of 4 pts in your group, you're missing the 'typical practice'

2) the authors claim in the abstract that pts were followed up to 3 years, but really are only analyzing one year. this is misleading.

3) why NOT include visit data from years 2 and 3? It may be that many of the elements that the authors are screening for aren't happening in the immediate postop period but become more relevant for years 2 and 3? To give an example, they mention that 0% had discussion of primary cancer screening. Most have not even had their surveillance colonoscopy within 1 year of completing treatment.

4) The authors make comments of over-use and under-use of visits... I imagine that if someone is getting seen 7-8 times in the year after a colorectal surgery it is for an active symptom management, not neccessarily 'overuse'... is there any insight into why there were more visits?

Minor:
1) Authors talk about confounding, but don't stratify their results by colon vs rectal or med-onc vs surg onc visits to see if there is a difference. I am surprised to see SO much 'survivorship' visit conducted by surgical oncology versus medical oncology.

2) who reviewed the medical records to see if the components were included? just 1 reviewer or multiple? how were disagreements handled?

Author Response

Thank you for taking the time to review our manuscript. We have provided our responses to your

insightful comments below.

  1. Comment: “Authors present a cohort of follow-up survival visits from their institution. During the 2.5 years of study, they include 48 pts, so <20 a year. This small volume per year (not the confounding, as they suggest in the limitations) is a major reason that raises concerns about the generalizability about the results.”

We have amended 4.1. Strengths and Limitations of the Discussion section, in order to state with more clarity, the significant limitations of our study (286-305).

  1. Comment: “The authors exclude 75% of their cancer patients -they claim that the reasons are ‘not reflective of typical practice’ but I would make the case if you are limiting to only 1 of 4 pts in your group, you’re missing the ‘typical practice’.”

We would like to make two important points, which we hope will address your comments regarding the exclusion of patients in our study, as depicted in Figure 1.

Firstly, Peter MacCallum Cancer Centre (Peter Mac) is a comprehensive cancer centre that is quaternary in nature. Patients are often referred to the colorectal cancer (CRC) surgical service to manage patients with recurrent rectal cancer and peritoneal metastatic disease. Compared to many CRC clinical services, Peter Mac sees few patients with de novo CRC, treated with curative intent. In our study, we wanted to identify a cohort of patients for which their treatment was completed in entirety at Peter Mac. This was to ensure that a date of surgery could be identified, that an index date could be elucidated, and that all follow-up visits were included. We have amended 2.1. Participants of the Methods section (89-95); and hope that this assists to explain how we arrived at our final population. We have also amended 4.1. Strengths and Limitations of the Discussion section, better clarifying the heterogeneity of the population as a limitation for the generalisability of our results (308-310).

Secondly, we would like to further explain the rationale behind the flow of participants through the study, as reflected in Figure 1. In the planning stages of our study, we had discussions with health information services, who provided us with the initial list of patients using health administrative data. A number of searches were attempted, which revealed that it was not readily possible to identify the date of curative resection for patients with non-metastatic colorectal cancer (CRC). Therefore, we asked to be provided with a list of patients with coding for CRC, but without any other additional cancer coding, or metastatic disease coding, who had an inpatient admission of any kind between July 2015 and January 2018. Each patient was then individually screened for inclusion into the study, as per the inclusion and exclusion criteria, therefore identifying which of those patients actually had curative surgery at Peter Mac during this time. Approximately 28% (n = 44) did not have curative surgery during this period – for example, their coded inpatient admission could have been for febrile neutropenia in the context of recent chemotherapy. The other reasons for exclusion are outlined in Figure 1, many of which reflect the inaccuracies of health administrative data, for example the 12 patients who had metastatic disease at diagnosis. As for the inclusion and exclusion criteria, these were informed by a systematic review of the follow-up of CRC survivors. Most studies in this area seek to exclude those with metastatic disease, or with genetic conditions that predispose to CRC. In addition, we had two clinical trials that ran during the study period, which mandated the follow-up visits and conditions of seven of our survivors, resulting in their exclusion. We do understand that our final population was small, but we are confident that they were reflective of typical practice. We have amended 2.1. Participants of the Methods section to explain that the initial inpatient admission was of any kind (76). We have also amended 4.1. Strengths and Limitations of the Discussion section to better reflect the inaccuracies of using coding data to identify populations (298-300).

  1. Comments: “Why NOT include visit data from years 2 and 3? It may be that many of the elements that the authors are screening aren’t happening in the immediate postop period but become more relevant for years 2 and 3? To give an example, they mention that 0% had discussion of primary cancer screening. Most have not even had their surveillance colonoscopy within 1 year of completing treatment” and “the authors claim in the abstract that pt were followed up to 3 years, but really are only analyzing one year. this is misleading.”

Given your comments regarding the reporting of years two and three, we have decided to include the data regarding the patterns of follow-up for these two years (3.3.2. Patterns of Follow- Up Visits in the Second Year, 192-198; and 3.3.3. Patterns of Follow-Up Visits in the Third Year, 199- 203). We believe that this information complements the data presented in Figure 2; demonstrating that follow-up continued to be consistent in year two but became more inconsistent in year three.

With regards to documented content of follow-up visits, our rationale behind limiting our analysis to the first follow-up year was that it contained all participants in our study (n = 48) and the most number of visits (n = 187). Our intention was to define a particular time period in the survivorship phase (i.e. in the year following treatment), in order to get a sense of what aspects of survivorship care are being addressed and/or discussed. Although we have preliminary data from years two and three, we believe that the low numbers of survivors and visits mean that few inferences can be made beyond the results presented in the first follow-up year.

We have reflected these changes in our abstract (18-19) and Figure 1 (136).

In our study-specific audit tool (Table A1), we consider surveillance colonoscopy as posttreatment surveillance. Fifty-six percent of survivors had relevant documentation of colonoscopy in the first year of follow-up. According to our study-specific audit tool, we considered discussions regarding screening for other primary cancers as relevant documentation. We have amended the wording in the abstract (22), in Figure 4 (223), and in our Results section (216), to avoid any confusion.

  1. Comment: “The authors make comments of over-use and under-use of visits… I imagine that if someone is getting seen 7-8 times in the year after a colorectal surgery it is for an active symptom management, not neccessarily ‘overuse’… is there any insight into why there were more visits?”

We agree that ‘overuse’ is potentially a misnomer; and we have amended this to ‘potential overuse’ (see Figure 3, 183). It is true that many of these extra visits could be for a specified purpose, say for a persistent physical symptom. However, as noted in Figure 2, some patients were receiving follow-up by multiple specialities, often not timed in concert, which appeared repetitive in nature. The point is that extra visits aren’t necessarily problematic, but they do perhaps reflect a problem, be it administrative, or that perhaps clinicians aren’t addressing an issue at hand. We have addressed this in the Discussion section of our manuscript (282-286).

  1. Comment: “Authors talk about confounding, but don’t stratify their results by colon vs rectal or med-onc vs surg onc visits to see if there is a difference. I am surprised to see SO much ‘survivorship’ visit conducted by surgical oncology versus medical oncology.”

In our original manuscript, we did not stratify by treatment but given comments by another reviewer, have included these results in our subsequent manuscript. The Discussion section of our manuscript has been amended in light of these comments (see 245-247, 250-251, 263-265),

including 4.1. Strengths and Limitations (see 306-310).

  1. Comment: “Who reviewed the medical records to see if the components were included? Just 1 reviewer or multiple? How were disagreements handled?”

We have amended 2.3.3. Data Collection and Handling of the Results section, to reflect that the study-specific audit tool was developed by all authors (114-115); and that review of the medical records was completed by the first author only (116-117).

Round 2

Reviewer 2 Report

The authors have expanded the methods, results and limitations in line with my previous concerns. this added transparency strengthens the manuscript.